# Physical-Chemical Characterization of Fruit Harvested at Different Maturity Stages of Grafted Yellow Pitahaya (*Selenicereus megalanthus* Haw.)

**DOI:** 10.3390/plants14020178

**Published:** 2025-01-10

**Authors:** Jessica Sanmiguel, Valdemar Andrade, Yadira Vargas-Tierras, Iván Samaniego, Fernando Paredes-Arcos, Wilson Vásquez-Castillo, William Viera-Arroyo

**Affiliations:** 1School of Agricultural and Environmental Sciences, Pontificia Universidad Católica del Ecuador Ibarra (PUCESI), Ibarra 100112, Ecuador; jpsanmiguel@pucesi.edu.ec (J.S.); vandrade@pucesi.edu.ec (V.A.); 2Instituto de Investigaciones Agropecuarias (INIAP), Estación Experimental Central de la Amazonía, Quito 170201, Ecuador; yadira.vargas@iniap.gob.ec (Y.V.-T.); fernando.paredes@iniap.gob.ec (F.P.-A.); 3Tumbaco Experimental Farm, Santa Catalina Research Site, National Institute of Agricultural Research (INIAP), Tumbaco 170902, Ecuador; ivan.samaniego@iniap.gob.ec (I.S.); william.viera@iniap.gob.ec (W.V.-A.); 4Ingeniería Agroindustrial, Universidad de Las Américas (UDLA), Redondel del Ciclista Vía a Nayón, Quito 170124, Ecuador

**Keywords:** grafting, fruit quality, bioactive compounds, antioxidant capacity

## Abstract

The physicochemical properties of fruits at different maturity stages using grafting technology are of great importance since grafting can alter the nutritional and functional parameters of the fruit. In this study, grafted yellow pitahaya (*Selenicereus megalanthus* Haw.) fruit, grown on live tutors, was evaluated from stages 0 to 5. The following response variables were recorded: fruit weight, diameter, and length; pulp weight with seed and peel; color; firmness; total soluble solids content; titratable acidity; pH; total flavonoid content; total polyphenol content; and antioxidant activity determined using FRAP and ABTS. The results show that fruits harvested from grafted plants have better physical characteristics such as fruit weight, diameter, and length. However, the total soluble solids content and titratable acidity were similar in fruits from grafted and ungrafted plants. The highest content of total polyphenols, flavonoids, and antioxidant activity determined by ABTS and FRAP were found in fruits at maturity stage 0, and the content decreased as the fruits ripened. A positive correlation was found between the total polyphenol content, total flavonoid content, and antioxidant capacity with protein content. The *S. megalanthus* grafting technique is a promising technology for sustainable production because it reduces pesticide use by combatting soil pathogens and not modifying fruit quality.

## 1. Introduction

In recent years, interest in consuming healthy foods has increased considerably, driven by the recognition of the benefits of a balanced diet, rich in nutrients and bioactive compounds [1,2]. Several studies support the positive relationship between the consumption of these compounds and the prevention of chronic diseases [3]. Fruits generate antioxidant compounds, such as betalains, which are responsible for their color and their polyphenol and flavonoid content, increasing their antioxidant capacity [4,5]. These antioxidants not only help to prevent cardiovascular disease, cancer, and neurodegenerative disorders but also help modulate inflammatory processes and improve metabolic health. Pitahaya, possessing a high concentration of these bioactive compounds, stands out as a functional food with a high potential to promote health and prevent chronic diseases [4,6]. In this context, the yellow pitahaya (*Selenicereus megalanthus* Haw.), also called yellow dragon fruit, is highly relevant for its nutritional, medicinal, and industrial properties [7]. It is important to mention that in addition to the content of soluble solids in the fruit, the importance of the medicinal properties (polyphenols, β-carotene and lycopene, and vitamins) is highly highlighted, which has caused the consumption of pitahaya to increase worldwide [8,9]. Pitahaya fruit is rich in amino acids, vitamins, polyphenols, sugars, pigments, and unsaturated fatty acids, in addition to the good taste, flavor, and color of the pulp [10,11,12]. Its seeds have a laxative effect and are also used for gastroenteritis [13]. Additionally, there are reports that cyanins contained in this fruit could control diabetes [14]. Many of these functional compounds increase as the fruit ripens [15].

These characteristics have allowed pitahaya fruit to enter several international markets, with the United States being the main destination market, with 84% of the total volume exported, followed by Canada, Colombia, and Singapore [16]. However, two major powers, China and Russia, have now been added as destination markets [17,18]. This increase in demand has led many countries, such as Vietnam, Malaysia, Mexico, Colombia, Costa Rica, Nicaragua, and Ecuador, to expand their production areas [19]. Similarly, Spain, Portugal, and other Mediterranean countries have also increased their production [20].

In Ecuador, there are 7216.73 hectares of pitahaya (yellow and red types) in production [21], cultivated in the provinces of Santo Domingo de los Tsáchilas, Manabí, Santa Elena, Los Ríos, Guayas, and El Oro on the coast; northwestern Pichincha, Bolívar, Imbabura, and Loja in central Ecuador; and Morona Santiago, Pastaza, Orellana, and Sucumbíos in the Ecuadorian Amazon [22]. Morona Santiago has the largest cultivated area (2400 ha) of yellow pitahaya [23], conventionally grown in monoculture, while in Orellana, yellow pitahaya is grown with more sustainable production technologies, such as the use of live tutors (*Erythrina* sp. and *Spondias mombin*) [24]. In research conducted by Vargas-Tierras et al., Castillo et al., and Vargas et al. [24,25,26], live stakes not only reduced costs and contributed nutrients to the soil but also provided shade to the pitahaya; according to studies conducted by Vargas et al. and Tomaz de Oliveira et al. [26,27], this plant needs 30% shade to carry out photosynthetic processes efficiently.

Climate change is affecting agriculture by causing abiotic stress to plants and restricting growth, development, and yield [28]. The use of rootstocks is a technological alternative that helps to mitigate environmental effects [29], improve physiological processes [28], reduce the juvenility period, accelerate production [30], and improve yield and fruit quality [31,32]. Also, the use of rootstocks can improve resistance to plant biotic factors such as pests [30] and diseases [33]. Therefore, the use of suitable rootstocks is a strategy that should be considered in integrated crop management [34].

Meanwhile, the excessive use of phytosanitary products in pitahaya monocultures has led to the development of resistance to pests, with one of these pests being nematodes [35], which is responsible for more than 80% of economic losses in crop production [36]. To reduce the use of pesticides, especially nematicides, tolerant rootstocks such as red pitahaya with white and red pulp of the genus *Hylocereus* are proposed [37]. Long et al. [36] studied *H. undatus,* and no infections by gall nematodes were observed, which suggests that this is a rootstock with tolerance against nematodes of the genus *Meloidogyne* spp. Therefore, grafting is presented as an effective technique that could reduce up to 90% of crop loss caused by this pest [38]. Yi-Lu et al. [39] reported that grafted yellow pitahaya grows more vigorously than the scion-propagated plant as a result of a strong root system of *H. undatus*; however, this technique could have effects on fruit quality. Adıgüzel et al. [40] pointed out that the use of grafting in fruit crops can alter nutritional quality by decreasing total soluble solids (TSS) content, varying pH, and affecting bioactive compounds such as total flavonoid content (TFC) and total polyphenol content (TPF). On the other hand, Abd-Elgawad [35] indicated that although vigor and disease tolerance increased in grafted watermelon plants, TSS content was not affected [36]. Forner-Giner et al. [41] pointed out that rootstocks play an interesting role in the juice content and bioactive compounds of citrus fruits, while Ziogas et al. [42] found that the rootstocks used for citrus did not influence the antioxidant capacity or the amount of juice in the fruit. In addition, Palacio Márquez et al. [43] concluded that grafting only affects fruit firmness, probably due to changes in physiological, anatomical, and metabolic aspects of the plant.

In general, the literature is lacking on this subject; therefore, this research aims to determine the influence of the rootstock on the quality (physical and chemical compounds) of pitahaya fruit, hypothesizing that the use of this cultural practice (grafting) does not affect fruit quality traits in the yellow pitahaya.

## 2. Results

### 2.1. Fruit and Pulp Color

The values of color coordinates, chromaticity, and hue angle are shown in Table 1. It can be seen that there were variations in the color parameters during the different maturity stages of the fruit development, both in the peel and the flesh of the fruit. However, the color difference (∆E) did not show much variation (more than 1 and less than 2); thus, a lower Delta E means there has been less color shift.

### 2.2. Morphometric Properties of the Fruit

Fruits harvested from grafted plants had higher weights (277.28 g) than fruits from ungrafted plants (273.33 g) at Stage 5; fruit growth was found to be exponential (Table 2). Fruit diameter and fruit length showed significant differences in the interaction between the grafting and the maturity stage. Fruits from grafted plants had the largest average diameter and length (81.50 and 105.94 mm, respectively) than those from ungrafted plants.

The analysis of pulp weight, including seed and flesh, was highly significant for the interaction between grafting and maturity stages. It was observed that fruits harvested on grafted and ungrafted plants showed similar pulp and peel weights. Firmness showed significant differences in the interaction. Fruits from grafted plants showed similar results to fruits from ungrafted plants; in addition, it was observed that fruits became less firm as they matured.

### 2.3. Proximal Analysis

The proximate analyses at the different maturity stages of grafted and ungrafted yellow pitahaya are shown in Table 3. The analysis of fruit pH revealed highly significant differences between fruit the interaction of grafting and maturity stage. The pH at the different stages decreased slightly for the fruit of ungrafted and grafted plants. TSS analysis showed significant differences in the interaction. TSS increased in both ungrafted and grafted plants during fruit development, but the values were similar between fruits from ungrafted and grafted plants at Stage 5. The analysis of titratable acidity also determined significant differences in the interaction. The acidity determined across the different maturity stages was variable; however, the highest percentage of acidity was observed in stage 2 of fruits harvested from ungrafted and grafted plants.

The protein analysis showed highly significant differences in the interaction between the grafting and maturity stages. The protein content of grafted pitahaya fruits was higher (18.73%) than that of fruits harvested from ungrafted plants (12.85%) at Stage 0; the same trend was also observed at Stage 5. The interaction also showed that the protein content was higher in fruits harvested from grafted plants than those from ungrafted plants.

### 2.4. Antioxidant Activity and Compounds

The univariate analysis of TPC and TFC found significant statistical differences between grafting and maturity stages. Antioxidant activity, measured by the ABST and FRAP methods, also revealed statistical differences. Fruits from ungrafted and grafted plants showed similar values for TPC, TFC, and antioxidant activity at Stage 5. Fruits from ungrafted plants presented higher values (72.55 µmol TE g^−1^) than those from grafted plants (63.59 µmol TE g^−1^). In general, the values for TPC, TFC, and antioxidant capacity (ABTS and FRAP) were higher at Stage 0 (Table 4), and they decreased from Stage 1 onwards.

### 2.5. Pearson’s Correlation Analysis

A Pearson’s correlation analysis for antioxidant compounds, TSS, protein, and pH was carried out for fruit from grafted and ungrafted plants (Figure 1). In both cases, a strong positive correlation was observed between protein and antioxidants (0.96 and 0.97, respectively), indicating that as antioxidant levels increase, so do protein levels. On the other hand, TSS showed a significant negative correlation with protein in both fruits from ungrafted and grafted plants (−0.91 and −0.89, respectively), as well as with antioxidant compounds, including TPC (−0.81 and −0.83, respectively) and TFC (−0.83 in both cases). This suggests that fruits with higher sugar content (i.e., sweeter fruits) tend to have lower levels of protein and antioxidants. In addition, TPC showed a perfect correlation (1) with antioxidant capacity measured by ABTS in both fruits from ungrafted and grafted plants, indicating that TPC is mainly responsible for antioxidant activity in fruits.

## 3. Discussion

Antioxidants are essential compounds that play a fundamental role in human health due to their antioxidant, antimicrobial, anti-inflammatory, and anticarcinogenic properties [44,45]. These compounds are widely distributed in fruits and vegetables, making them an important source of human nutrition [44]. However, it is crucial to perform a thorough analysis to ensure that the antioxidants present are directly bioavailable for human consumption [46]. Several factors can influence the concentration and quality of antioxidants in crops, and one of the most prominent is the use of rootstocks [47]. The implementation of grafting can cause alterations in the chemical composition of plants, modifying both the profile and the amount of antioxidants [48]. These variations depend on the interaction between the rootstock and the scion, which affects the plant’s ability to synthesize and accumulate antioxidants optimally. In addition, factors such as sugar and phenolic compound contents are essential in order to evaluate the fruit quality index [49], which underlines the importance of choosing rootstocks appropriately to maximize the nutritional benefits of the cultivars.

### 3.1. Color Parameters

The color of both the peel and pulp is a key indicator of the maturity process of the yellow pitahaya. Although color changes between the first and final stages are perceptible to the naked eye during maturity, the changes in the middle stages are not easy to differentiate due to the high similarity of the observed shades [50]. To overcome this limitation, current measurement methods, such as the CIELab system, produce more precise and objective data. In this research, variability in peel color was evidenced depending on the maturity stage. In the immature stages, the fruits were bluish-green, while in the mature stages, greenish-yellow tones predominated. Hinojosa-Gómez et al. [51] reported values of 2.2 (a*), 55 (b*), and 83.5 (L*) for yellow pitahaya fruits at commercial maturity, values that differ from the results of this study. However, the tonality of the peel coloration is influenced by the plant’s nutritional and climate conditions, such as hours of daylight [52,53]. It has been mentioned that overripe fruits had a reduced yellow content [54] and that the values of a* and C* increase but H° decreases [55,56], which are criteria that coincide with the results of this study.

### 3.2. Effect of Grafting on Fruit Morphometric Properties

The weight of grafted pitahaya fruit increased across the five stages as the fruit matured, reaching a weight of 277.28 g, while the weight of ungrafted yellow pitahaya fruit reached a weight of 273.33 g. The latter value is similar to the weights reported by Morillo-Coronado et al. [57], where fruits from different genotypes of ungrafted pitahaya weighed 265.60 g to 278.50 g. Yi-Lu et al. [39] also mentioned that in sweet cherries, the rootstock does not influence the weight, diameter, or length of the fruit [58].

The TSS values at Stage 5 from the fruit of ungrafted plants reached a value of 20.07 °Brix. This value coincides with values (20.74 °Brix) obtained by Verona Ruiz et al. [10], while Sotomayor et al. [59] found that *H. megalanthus* has a percentage of TSS of 16 to 17%, i.e., 20.74 °Brix. In addition, TSS in fruits from ungrafted plants at Stage 5 in our study was higher or similar to the content determined by Morillo-Coronado et al. [57] and Vasquez et al. [60], who reported values of 15.85 and 17.90 to 20.10 °Brix, respectively, in ripe ungrafted pitahaya. TSS content obtained in this study (plants grown on *S. mombin* stakes) was lower than the contents achieved in pitahaya fruit grown in full sunlight. Tomaz de Oliveira et al. [61] asserted that TSS is higher in pitahaya fruit grown in open fields because of the greater exposure to sunlight [62]. In studies conducted by Yi-Lu et al. [39], it was found that, unlike what is commonly observed in sunnier seasons, pitahaya can reach a value of 20.80 °Brix even during winter. This finding contrasts with the usual trend in which TSS tends to increase with solar exposure. The results suggest that, although sunlight is a key factor for sugar content, other climatic factors, such as temperature and humidity in winter, could influence the accumulation of sugars in pitahaya, allowing it to maintain high levels of sweetness.

The pH and TA, in the different maturity stages, showed values similar to those reported by Grisales et al. [63]. Meanwhile, Enciso et al. [64] observed that in *H. undatus* at three stages of maturity (initial, medium, and complete), the titratable acidity values were higher compared to those of *S. megalanthus* (0.92, 0.76, and 0.63). Pulp firmness decreased as the fruit matured, reaching a value of 4.11 N cm^−2^ at maturity (Stage 5) for fruit harvested from ungrafted plants. This behavior has been corroborated in research conducted by Vera et al. [65], where a firmness of 4.89 N cm^−2^ was reported at the mature stage. In addition, a similar tendency was observed in *H. undatus*, which presents a firmness of 4.20 N cm^−2^ [64].

### 3.3. Total Polyphenol Content, Total Flavonoid Content, and Antioxidant Activity

TPC in fruits is crucial both from a nutritional and commercial point of view [46]. In pitahaya, TPC stands out for its high antioxidant activity, contributing not only to the fruit’s sensory and organoleptic characteristics but also to its benefits for human health [63]. Regular consumption of antioxidant-rich fruits, such as pitahaya, has been associated with a reduced risk of neurodegenerative diseases and premature aging by counteracting the harmful effects of oxidative stress in the body [66]. This health improvement comes from the antioxidant power of polyphenols, which neutralize free radicals, minimizing cell damage and promoting overall well-being. The TPC in the yellow pitahaya fruit pulp was higher at Stage 0, dropping drastically thereafter at Stage 5. In *H. undatus*, Campozano et al. [67] discovered that as the maturity stage increased, the TPC decreased from 45.40 to 40.38 mg GAE/100 mL. Lupuche et al. [68] stated that the TPC in *H. megalanthus* was 12% higher in the pulp than in the peel. On the other hand, the TPC content in ripe red pitahaya fruits has been reported to be between 16.66 and 17.11 mg GAE g^−1^ FW [19].

TFC in the pulp of the fruit at Stage 0 was 18.34 mg CatEq/g FW for fruit coming from grafted plants, while by Stage 5, the flavonoids had decreased to 0.72 mg CatEq/g FW. This same tendency was reflected in ungrafted pitahaya. Guevara et al. [69] determined that TFC in yellow pitahaya presented an average of 3.57 mg CatEq/g FW in ripe fruits. In studies carried out on ripe red pitahayas by Singh et al. [70], the TPC was 144.8 mg CatEq/g FW on average in immature stages, dropping to 72.69 mg CatEq/g FW. Xie et al. [71] also pointed out that TPC and TFC are at their highest during the initial stage of fruit development (immature fruits), gradually decreasing as maturity progresses to their lowest point at the time of harvest. This same behavior was reported by Erazo-Lara et al. [72] for yellow pitahaya, where TPC and TFC contents were 6 and 20 mg g^−1^, respectively, at Stage 1 (immature fruit). Some studies indicate that the TFC is higher in the peel than in the pulp, e.g., Tendafilova et al. [73]. Other studies mention that TFC varies depending on crop conditions, specific maturity at the time of harvest, and genetic variations [19,70]. The antioxidant capacity evaluated by ABTS and FRAP methods showed that the content decreased as the fruits matured in both grafted and ungrafted pitahayas [70].

This pattern has also been observed in previous investigations, such as those by Campozano et al. [67] on *H. undatus*, where reductions in antioxidant capacity were recorded during Stages 3 and 4 of maturity (34.97 and 32.43 mg TEAC/100 mL, respectively). Zapata et al. [74] also confirmed that the antioxidant capacity is higher in unripe stages of *Psidium araca* fruit (6679.92 and 623.98 μmol of Trolox/100 g of fresh pulp by ABTS and FRAP methods, respectively). Additionally, studies by Castro-Enríquez et al. [75], in different ecotypes of red pitahaya, showed higher levels of antioxidant capacity according to the ABTS method, attributing these variations to ultrafiltration of the extract. Lupuche et al. [68] found that in both yellow and red pitahaya ecotypes, the antioxidant values measured by ABTS were relatively high at their optimum ripening point (579.46 and 565.62 µmol TE g^−1^ for yellow and red pitahaya, respectively).

The results show a significant and positive correlation between TPC, TFC, and the antioxidants evaluated (ABTS and FRAP), indicating that the highest levels are found in the immature stage of the fruit and decrease as it matures. This finding was corroborated by Erazo-Lara et al. [72], where the same behavior was reported in yellow pitahaya grown in the southern Ecuadorian Amazon. Thus, it can be concluded that both TPC and TFC are positively correlated with antioxidant activity, suggesting that higher levels of TPC and TFC correspond to higher antioxidant activity. However, the specific assessment of antioxidant power depends on the method used and the measurement parameters.

## 4. Materials and Methods

### 4.1. Location

This research was conducted at the Food Quality Laboratory of the Central Amazon Experimental Station (EECA), belonging to the National Institute of Agricultural Research (INIAP), located in San Carlos parish, La Joya de los Sachas canton, Orellana province, Ecuador. The sampled fruits were collected from the PitaCastro farm, located in La Joya de los Sachas, Orellana province, with coordinates 20°27″ N latitude and 87°40″ E longitude (Figure 2) and an altitude of 282 m above sea level. This area has a humid subtropical climate, with abundant rainfall throughout the year (4162 mm annually), an average annual temperature of 23.6 °C, and a relative humidity of 91.5% [76].

### 4.2. Preparation of Plant Material

Yellow pitahaya plants *(S. megalanthus*) grafted onto red pitahaya (*H. undatus*) were used alongside ungrafted plants. Sampling was carried out during the season of least precipitation (236 mm) [77], in the months of September and November [78]. The soil physical analysis of the experimental plot determined that the soil had a sandy loam texture, moderately acidic pH (5.43), and an organic matter content of 4.67%. Fertilization was carried out according to the crop requirements based on the soil analysis [24]. Cultural tasks, such as weed control; the pruning of live stakes to regulate shade (30–60%); and maintenance pruning to eliminate broken, diseased, and intersecting branches, were performed every 30 days.

The study began with monitoring the phenological cycle of the fruit, which lasted 127 days from the formation of the floral bud until the fruit reached stage 0. At this stage, the spines of the fruit acquire a red color, and the fruit remains in this state for 37 days. Subsequently, the thorns change to a brown tone, marking the beginning of stage 1, in which the fruit remains for 10 days. From this point on, the changes corresponding to states 2, 3, 4, and 5 occur at intervals of 4 days each. For the determination of physical, chemical, bioactive, and antioxidant compounds, 40 fruits were harvested in state 0, 20 fruits in state 1, and 18 fruits from state 2 to 5. The number of fruits varied based on the size of the fruit and its pulp content. For fruit harvesting, a nylon brush and gloves were used to remove the thorns from the harvested fruit. The bags were labeled, and the fruits were placed in plastic crates to be taken to the laboratory, where the physicochemical parameters were determined. For the remaining fruits, the peel and pulp were separated; each fraction was frozen at −15 °C, then lyophilized, and an analysis of antioxidant and bioactive compounds was performed.

The description of the stages of maturity of the yellow pitahaya fruit was based on the time elapsed between each stage and visual characteristics (Figure 3). In state 0, the fruit had a dark green color with rigid, reddish spines. In state 1, the tonality changed to light green, while the spines remained hard but acquired a brown color. During stage 2, yellow spots appeared, the filling of the bracts began, and the spines remained rigid and brown. In stage 3, the fruit acquired a yellowish-green tone; the bracts showed a green surface, which continued to fill and began to separate further from each other. In stage 4, the color of the fruit became predominantly yellow, with slight green tones in the bracts, which were already completely full and separated. Finally, in stage 5, the fruit reached a uniform yellow color, with the tips of the bracts green, while the spines appeared completely dry and aged [79].

### 4.3. Color Evaluation

The color was evaluated according to the maturity stage. Color measurements according to the CIELAB system (L* a* b*) were performed on both whole fruit and pulp using a Konica Minolta Chroma Meter CR-400 tristimulus (Konica Minolta Sensing Americas, Ramsey, NJ, USA). Chromatic properties were defined using the CIE (Commission internationale de l’éclairage) L* a* b* color system, where L* represents lightness, a* indicates the red/green component, and b* blue/yellow [46]. In addition, the Delta E (∆E) value (color difference), hue angle (H), and chroma (C*) were calculated from the values of L*, a*, and b* using the following formulas: ∆E = [(L_2_ − L_1_)^2^ + (a_2_ − a_1_)^2^ + (b_2_ − b_1_)^2^]^1/2^; H = tan^−1^ (b/a); C* = (a^2^ b^2^)^1/2^.

### 4.4. Evaluation of the Morphometric Properties of S. megalanthus

The weight of the fruit, pulp with seeds, and peel was determined using a Citizen SCALE CG 4102C analytical balance (Gardena, CA, USA). The diameter and length of the fruits were measured with a digital electronic caliper (CD-6, Mitutoyo, Kawasaki, Japan). Pulp firmness was evaluated in the central region of the fruits using a digital penetrometer (GY-4, BIOBASE, Shandong, China) with an 8 mm tip, and the results were recorded in Newtons (N/cm^2^).

### 4.5. Proximate Analysis

The edible part of the fruit (pulp with seeds) was ground in a mortar and pestle until a homogeneous paste was obtained, which was used to quantify the TSS using a digital refractometer (ATAGO, Tokyo, Japan). The results were expressed in °Brix. Additionally, the pH was measured with a portable digital pH meter (model PT-380, Boeco, Hamburg, Germany). For the titratable acidity analysis, a sample was crushed and homogenized at a 1:10 ratio (10 g of pulp diluted in 100 mL of distilled water), expressed in grams of citric acid per 100 mL. A 20 mL sample was taken and titrated with 0.1 N sodium hydroxide until reaching the endpoint at pH 8.2 [80]. Protein content (g/100 g) was measured following the standard procedures established by the Association of Official Analytical Chemists (AOAC) [81]. The Kjeldahl method was used, employing 0.5 g of dehydrated sample, which was digested at 400 °C, followed by distillation and titration with 0.3 N sulfuric acid, using an automatic distiller (Vapodest 45s, Königswinter, Germany) and a mixed indicator, and by observing the color change.

### 4.6. Bioactive Compounds

#### 4.6.1. Total Polyphenol Content

TPC was extracted from defatted pitahaya powder by continuous agitation with a 70% methanol aqueous solution for 45 min. A small sample of the obtained extract was used to perform a colorimetric reaction using the Folin–Ciocalteu reagent. The absorbance of the blue chromophore was measured at 760 nm using a UV-VIS spectrophotometer (model 2600, Shimadzu, Kyoto, Japan). TPC was expressed as mg of Gallic Acid Equivalents per 100 g of fresh weight (mg GAE/100 g FW) [46].

#### 4.6.2. Total Flavonoid Content

The estimation of TFC was measured using a UV-VIS spectrophotometer (model 2600, Shimadzu, Kyoto, Japan) using the method by Singh et al. [70]. The sample was homogenized in 10 mL of 80% methanol. Then, a total aliquot of 500 µL of aluminum chloride, ethanol, potassium acetate, and distilled water was added. The absorbance was measured in the pink chromophore at a wavelength of 500 nm using a UV-VIS spectrophotometer (model 2600, Shimadzu, Kyoto, Japan), and the TFC was expressed as mg catechin per 100 g fresh weight (mg CatEq/100 g FW) [46].

### 4.7. Determination of Antioxidant Activity

#### 4.7.1. Antioxidant Capacity by the ABTS Method

Antioxidant activity (AA) was determined using the 2,2-azino-bis(3-ethylbenzothiazoline-6-sulfonic) acid (ABTS) radical bleaching method [59]. An ABTS solution (7 mM) and potassium persulfate (2.45 mM) were prepared in a 1:1 (*v*/*v*) ratio. The absorbance of this solution was measured the next day and diluted with a phosphate buffer to achieve an absorbance of 1.1 × 0.01 at 734 nm. Then, 3.8 mL of this ABTS working solution was added to 15 mL test tubes containing 200 µL of the sample, and the solution was allowed to rest for 45 min. The absorbance was measured at 734 nm using a UV-VIS spectrophotometer (model 2600, Shimadzu, Kyoto, Japan). The AA was determined by interpolating the absorbance on a previously developed calibration curve using Trolox as the standard (0.800). The results were expressed as Trolox equivalents in moles per gram of dry sample (mol TE g⁻^1^ DW) [46,82].

#### 4.7.2. Antioxidant Capacity by the FRAP Method

A calibration curve was created using Trolox as the standard and a pH 6.6 buffer, covering concentrations from 0 to 600 ppm from an initial solution of 2000 µmol/L. Next, 1 mL of the sample was mixed with 2.5 mL of pH 6.6 buffer and 2.5 mL of 1% potassium ferricyanide, agitated, and heated at 50 °C for 20 min. Then, 2.5 mL of 10% trichloroacetic acid, 2.5 mL of water, and 0.5 mL of 1% ferric chloride were added, and it was agitated again. The solution was kept in the dark for 30 min, and its absorbance was measured at 700 nm using a UV-VIS spectrophotometer (model 2600, Shimadzu, Kyoto, Japan) [82].

### 4.8. Statistical Analysis

In this research, a statistical analysis was performed using three replicates, totaling six experimental units. The net plot consisted of 25 plants. Three fruits were sampled per treatment and repetition at different maturity stages, for a total of 264 fruits, of which 132 were used for the determination of morphometric and proximate parameters, while the remaining fruits were used for the determination of antioxidant and bioactive compounds. The statistical analysis was conducted using the R software (version 4.2.3). An analysis of variance (ANOVA), as well as the Tukey test with a significance level of 0.01 to identify significant differences between means, were used. The correlation analysis was conducted using Pearson’s correlation coefficient, which is shown in a heatmap.

## 5. Conclusions

The aim of the present study was to determine the impact of rootstock on the physicochemical quality of pitahaya fruit. The analysis of the physical characteristics of the fruits at Stage 5 established that fruits from grafted plants exhibited better weight, length, and diameter values; however, other traits such as firmness, pulp, and peel weight showed similar results for both fruits from ungrafted and grafted plants. Regarding the proximal analysis, pH, SSC, and titratable acidity were similar for fruits harvested at stage 5 from ungrafted and grafted plants; only protein content was higher in fruits from grafted plants. Maintaining a stable SSC is good because this is the main parameter appreciated by the consumers and the agroindustry.

TPC, TFC, and antioxidant activity were higher in the initial fruit state (Stage 0) and decreased as the fruit ripened; however, these parameters were similar between the fruits harvested at Stage 5 from ungrafted and grafted plants. Consequently, the results of this study indicated that the rootstock would not produce changes in the fruit’s chemical characteristics and biocompound content, which is positive in terms of fruit quality traits.

It can be mentioned that the grafting technique in the fruit crop of *S. megalanthus* is a promising technology for sustainable production, as it helps minimize pesticide use for controlling *Meloidogyne incognita*, supports the plant during sudden climate changes, and improves certain fruit parameters such as fruit size and protein content, while maintaining the other main fruit quality traits (SSC, TPC, and TFC) similar to fruits harvested from ungrafted plants.

Finally, since there are many different factors that affect fruit quality fruit and consequently the profitability of cultivation, it is necessary to conduct long-term studies to confirm the results in different productive seasons, as well as carry out research to find the best scion/rootstock combination, within the constraints imposed by local climate and economic conditions.

## Figures and Tables

**Figure 1 plants-14-00178-f001:**
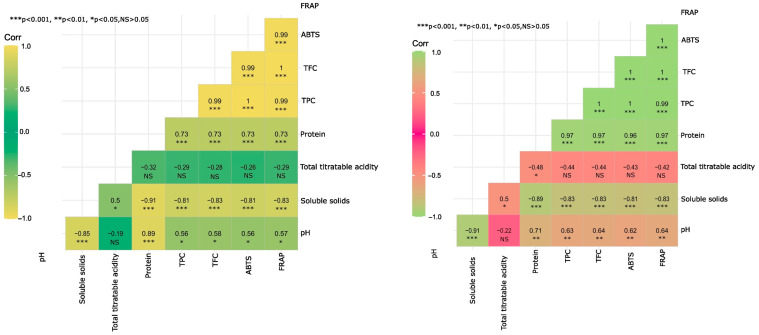
Heat maps showing Pearson’s correlation coefficients between antioxidant properties (ABTS and FRAP; µmol TE g^−1^), total polyphenol content (mg GAE g^−1^), total flavonoid content (mg catechin g^−1^), and proximate components: protein, pH, titratable acidity, and total soluble solids. (**left**) Pitahaya fruit from grafted plants; (**right**) Pitahaya fruit from grafted plants.

**Figure 2 plants-14-00178-f002:**
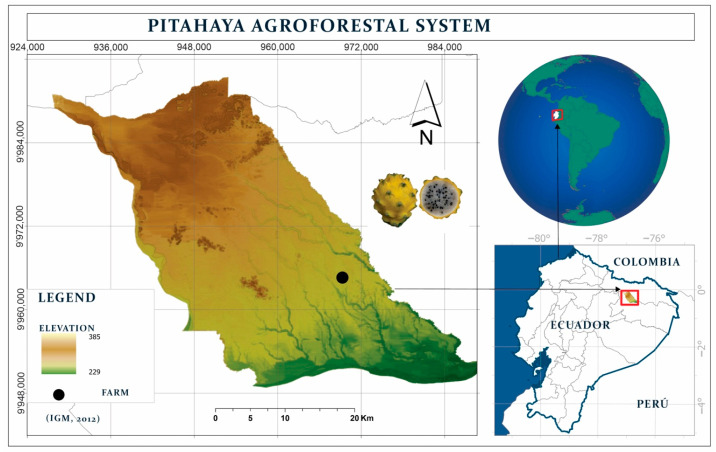
Location of the pitahaya agroforestry system in La Joya de los Sachas canton, Orellana province.

**Figure 3 plants-14-00178-f003:**
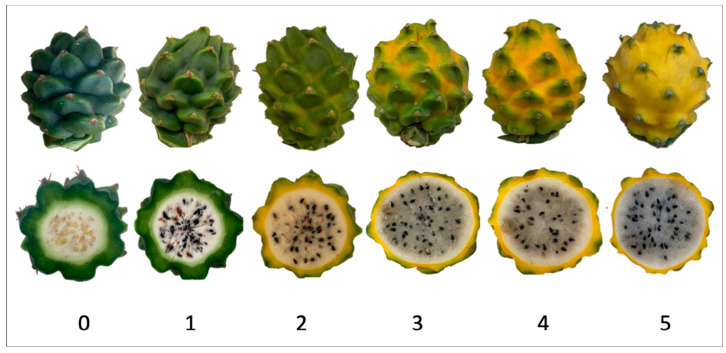
Color chart of the different maturity stages of *S. megalanthus* fruits.

**Table 1 plants-14-00178-t001:** Color parameters of fruit and flesh at different maturity stages of *Selenicereus megalanthus.*

Grafting	Maturity Stage	L* Lightness	a* (+Red, −Green)	b* (+Yellow, −Blue)	C* Chroma	H° Hue	∆E Delta E
Ungrafted Grafted	0 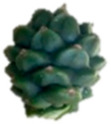	31.67 ± 1.01 de 32.10 ± 3.69 cde	−5.28 ± 0.29 e −5.36 ± 0.76 e	10.93 ± 0.83 f 11.63 ± 1.53 ef	12.14 ± 0.87 e 12.81 ± 1.70 e	205.84 ± 0.61 a 204.69 ± 0.61 a	1.33
Ungrafted Grafted	1 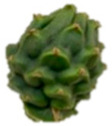	30.74 ± 3.36 e 27.73 ± 0.63 e	−3.91 ± 0.72 e −3.43 ± 0.22 e	7.75 ± 1.21 f 6.47 ± 0.28 f	8.68 ± 1.41 ef 7.33 ± 0.16 f	206.73 ± 0.98 a 207.96 ± 2.50 a	1.89
Ungrafted Grafted	2 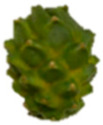	38.33 ± 1.43 cd 38.78 ± 1.77 c	−6.05 ± 1.06 e −5.30 ± 1.00 e	18.63 ± 2.49 de 20.56 ± 1.86 d	19.64 ± 2.12 d 21.26 ± 1.69 d	105.79 ± 4.72 b 88.85 ± 15.24 c	1.71
Ungrafted Grafted	3 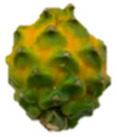	53.43 ± 0.78 ab 48.19 ± 2.77 b	7.61 ± 0.7 c 3.35 ± 2.23 d	42.75 ± 1.87 b 35.47 ± 4.19 c	43.42 ± 1.93 b 35.66 ± 4.38 c	79.91 ± 0.71 de 84.82 ± 2.87 cd	2.66
Ungrafted Grafted	4 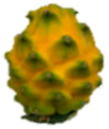	59.19 ± 2.11 a 55.14 ± 0.70 ab	11.84 ± 1.64 ab 8.82 ± 0.72 bc	50.54 ± 2.92 a 42.13 ± 3.39 bc	51.91 ± 3.19 a 43.05 ± 3.41 b	76.85 ± 1.14 e 78.16 ± 0.85 de	2.58
Ungrafted Grafted	5 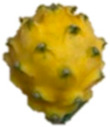	59.87 ± 4.49 a 58.70 ± 1.94 a	12.68 ± 0.69 a 8.55 ± 1.72 bc	48.58 ± 3.62 ab 45.00 ± 1.69 ab	50.23 ± 3.34 a 45.82 ± 1.98 b	75.30 ± 1.80 e 79.29 ± 1.69 de	2.24
Ungrafted Grafted	0 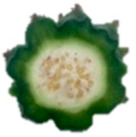	79.43 ± 0.33 a 75.55 ± 1.54 a	2.60 ± 0.33 a 3.35 ± 0.58 a	24.64 ± 1.76 a 23.52 ± 0.93 a	24.78 ± 1.78 c 23.76 ± 1.00 c	83.99 ± 0.48 ab 81.93 ± 1.08 b	1.97
Ungrafted Grafted	1 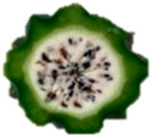	70.18 ± 4.60 a 69.55 ± 8.29 a	−0.24 ± 0.20 b −0.07 ± 0.49 b	11.29 ± 1.68 b 12.69 ± 1.02 b	11.29 ± 1.68 b 12.69 ± 1.02 b	91.24 ± 0.95 a 90.21 ± 2.22 a	1.55
Ungrafted Grafted	2 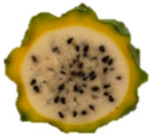	44.62 ± 4.52 b 42.39 ± 6.26 b	0.24 ± 0.42 b 0.01 ± 0.19 b	5.90 ± 0.74 c 6.09 ± 1.04 c	5.91 ± 0.74 a 6.09 ± 1.04 a	87.67 ± 4.06 ab 89.76 ± 1.92 a	1.55
Ungrafted Grafted	3 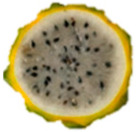	37.24 ± 3.84 b 41.50 ± 2.08 b	0.30 ± 0.86 b 0.53 ± 0.23 b	4.35 ± 0.76 c 6.50 ± 1.09 c	4.41 ± 0.79 a 6.52 ± 1.09 a	86.16 ± 10.51 ab 85.33 ± 1.86 ab	2.00
Ungrafted Grafted	4 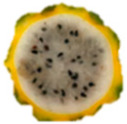	38.71 ± 4.88 b 39.19 ± 2.94 b	0.39 ± 0.80 b 0.11 ± 0.32 b	4.50 ± 1.10 c 5.23 ± 1.14 c	4.56 ± 1.15 a 5.24 ± 1.15 a	86.09 ± 9.21 ab 89.16 ± 3.05 ab	1.44
Ungrafted Grafted	5 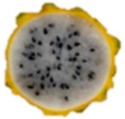	39.59 ± 1.06 b 35.26 ± 1.72 b	0.25 ± 0.18 b 0.28 ± 0.16 b	4.94 ± 0.47 c 3.70 ± 0.66 c	4.95 ± 0.47 a 3.71 ± 0.65 a	87.05 ± 2.24 ab 85.51 ± 3.02 ab	1.84

Different letters in each parameter indicate significant differences between grafting and maturity stages (*p* = 0.05).

**Table 2 plants-14-00178-t002:** Effect of grafting and maturity stage on morphometric properties of the fruit of *Selenicereus megalanthus.*

Grafting	Maturity Stage	Fruit Weight (g)	Fruit Diameter (mm)	Fruit Length (mm)	Pulp Weight (g)	Peel Weight (g)	Fruit Firmness (N)
	0	154.07 ± 3.33 g	60.60 ± 0.09 e	82.01 ± 3.64 f	44.92 ± 2.51 g	162.18 ± 18.82 b	18.98 ± 0.39 a
	1	200.11 ± 15.77 f	61.71 ± 1.62 e	88.61 ± 3.18 def	59.00 ± 0.43 f	118.04 ± 1.92 cde	11.02 ± 0.22 b
Ungrafted	2	220.60 ± 0.60 de	64.11 ± 1.69 de	92.48 ± 1.3 cde	103.94 ± 3.45 e	107.43 ± 4.73 def	5.27 ± 0.29 cd
	3	234.18 ± 2.79 cd	67.41 ± 2.10 cd	95.76 ± 0.87 cd	123.01 ± 2.59 d	95.16 ± 5.96 efg	4.34 ± 0.02 de
	4	259.09 ± 6.29 b	68.22 ± 1.16 cd	96.69 ± 0.45 bc	148.95 ± 5.3 b	86.00 ± 1.03 fg	4.25 ± 0.04 de
	5	273.33 ± 0.86 ab	71.56 ± 0.80 c	98.90 ± 0.68 abc	164.87 ± 4.22 a	78.62 ± 7.58 g	4.11 ± 0.05 de
	0	185.58 ± 1.30 1 f	64.69 ± 1.72 de	81.70 ± 1.60 f	39.34 ± 4.48 g	195.42 ± 16.53 a	19.29 ± 1.01 a
	1	216.22 ± 4.34 e	67.93 ± 2.13 cd	86.86 ± 6.16 ef	49.69 ± 8.66 fg	139.12 ± 2.84 bc	11.24 ± 0.48 b
Grafted	2	221.93 ± 0.44 de	71.39 ± 1.96 c	94.75 ± 0.84 cd	94.12 ± 3.14 e	125.43 ± 8.41 cd	5.73 ± 0.82 c
	3	241.85 ± 2.58 c	76.57 ± 2.17 b	96.34 ± 0.71 bc	117.38 ± 1.69 d	108.03 ± 6.90 def	4.43 ± 0.55 cde
	4	265.05 ± 3.29 ab	79.40 ± 0.64 ab	103.32 ± 2.04 ab	136.37 ± 1.70 c	100.05 ± 8.10 defg	3.58 ± 0.17 e
	5	277.28 ± 1.72 a	81.50 ± 1.33 a	105.94 ± 2.11 a	163.07 ± 4.84 a	89.60 ± 1.94 fg	3.33 ± 0.21 e

Different letters in each parameter indicate significant differences between grafting and maturity stages (*p* = 0.05).

**Table 3 plants-14-00178-t003:** Proximal analysis of *Selenicereus megalanthus* according to grafting and maturity stage.

Grafting	Maturity Stage	pH	Soluble Solids (°Brix)	% Titratable Acidity	Protein (g/100 g)
	0	4.88 ± 0.01 a	3.60 ± 0.10 g	0.24 ± 0.02 cd	12.85 ± 0.66 b
	1	4.82 ± 0.01 a	9.47 ± 0.23 f	0.19 ± 0.00 d	11.71 ± 1.21 bc
Ungrafted	2	4.68 ± 0.05 b	17.8 ± 0.50 de	0.41 ± 0.05 ab	10.58 ± 0.52 cde
	3	4.35 ± 0.06 d	19.07 ± 0.45 abc	0.38 ± 0.03 b	9.39 ± 0.30 de
	4	4.18 ± 0.03 e	19.82 ± 0.20 a	0.35 ± 0.02 bc	9.22 ± 0.29 e
	5	4.13 ± 0.04 e	20.07 ± 0.06 a	0.20 ± 0.02 d	8.87 ± 0.36 e
	0	4.93 ± 0.03 a	3.33 ± 0.31 g	0.22 ± 0.01 d	18.73 ± 0.99 a
	1	4.84 ± 0.01 a	9.33 ± 1.11 f	0.23 ± 0.01 cd	12.57 ± 0.64 b
Grafted	2	4.55 ± 0.04 c	16.80 ± 0.30 e	0.50 ± 0.10 a	11.50 ± 0.41 bc
	3	4.42 ± 0.06 d	18.31 ± 0.10 cd	0.41 ± 0.06 ab	11.52 ± 0.24 bc
	4	4.35 ± 0.04 d	18.55 ± 0.13 bcd	0.36 ± 0.01 b	11.10 ± 0.11 bcd
	5	4.11 ± 0.04 e	19.57 ± 0.15 ab	0.23 ± 0.02 d	11.45 ± 0.38 bc

Different letters in each parameter indicate significant differences in maturity stages (*p* = 0.05).

**Table 4 plants-14-00178-t004:** Bioactive compounds and antioxidant activity in *Selenicereus megalanthus* according to grafting and maturity stages.

Grafting	Maturity Stage	Polyphenols (mg GAE g^−1^)	Flavonoids (mg catechins g^−1^)	ABTS (µmol TE g^−1^)	FRAP (µmol TE g^−1^)
	0	33.23 ± 7.29 a	20.58 ± 1.29 a	554.21 ± 98.08 a	311.99 ± 47.17 a
	1	3.35 ± 0.28 b	1.28 ± 0.29 c	55.72 ± 0.33 b	33.90 ± 3.19 c
Ungrafted	2	2.81 ± 0.21 b	1.35 ± 0.19 c	72.26 ± 3.98 b	24.66 ± 3.23 c
	3	2.24 ± 0.29 b	0.66 ± 0.10 c	40.21 ± 2.19 b	20.96 ± 1.63 c
	4	2.38 ± 0.23 b	0.55 ± 0.14 c	49.59 ± 1.00 b	21.20 ± 2.96 c
	5	2.87 ± 0.29 b	0.85 ± 0.09 c	57.7 ± 1.05 b	22.57 ± 1.81 c
	0	28.67 ± 2.15 a	18.34 ± 0.68 b	498.88 ± 37.67 a	255.28 ± 8.89 b
	1	3.23 ± 0.20 b	1.20 ± 0.11 c	48.84 ± 7.69 b	28.99 ± 4.72 c
Grafted	2	2.31 ± 0.34 b	0.60 ± 0.13 c	41.60 ± 5.74 b	23.14 ± 4.89 c
	3	2.66 ± 0.24 b	0.76 ± 0.05 c	44.83 ± 3.77 b	26.61 ± 1.54 c
	4	2.82 ± 0.32 b	0.84 ± 0.31 c	56.65 ± 1.2 b	25.20 ± 2.86 c
	5	2.74 ± 0.07 b	0.72 ± 0.07 c	53.67 ± 7.67 b	22.29 ± 0.93 c

Different letters in each parameter indicate significant differences between grafting and maturity stages (*p* = 0.05).

## Data Availability

Data are contained within the article.

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
