# Peer review of "Physical-Chemical Characterization of Fruit Harvested at Different Maturity Stages of Grafted Yellow Pitahaya (Selenicereus megalanthus Haw.)"

_plants, 2025, doi:10.3390/plants14020178_

Round 1
Reviewer 1 Report
Comments and Suggestions for Authors
Dear Autors,
The manuscript “Integral exploration of the functional and antioxidant proper-ties of grafted yellow pitahaya (Selenicereus megalanthus Haw.)” is an interesting paper about the the effects of rootstock the physico-chemical and nutritional quality of yellow pitahaya grafted onto red pitahaya at five harvest times.
The overall level of the paper is good, it is well written and some important considerations are highlighted. This paper has a potential to be accepted, but some important points have to be clarified or fixed.
1 The introduction could be strengthened by providing more detailed information on the quality and nutritional importance of pitahaya fruit Additionally, a brief overview of effects of rootstock on internal quality parameters in several species would set the stage for the study's approach.
2 The authors should verify that all statistical tests are appropriately applied and that the results are accurately interpreted. The study focuses on statistical differences, however, the authors should better explain the harvest period/rootstock interaction. The results indicate rootstock p-values and harvest p-values.
3 The authors highlight the differences in quality parameters between grafted and non-grafted plants and/or different ripening stages. Could they hypothesize/report what explains these differences?
4 The studied plants were cultivated in one field collection or in the same pedo-climatic environment? I think it is also important to go more in details on how the cultivation was done, with details on the soil parameters, and fertilization and irrigation regimes.
5 I think it is important that the authors indicate the fruit harvesting period and the temporal sequence of the sampling.
6 A very important aspect is the preparation of the samples. How much time passed between the harvest and the processing of these fruits? How many fruits were collected and used in subsequent analyses?
Since there are many different factors which affect the qualities of a fruit and consequently the profitability of a cultivation, it is necessary to conduct long-term studies to find the best scion/rootstock combination, within the constraints imposed by local climate and economic conditions. I would therefore suggest that the authors repeat the experiments to confirm the results obtained in this research.
In general terms, the rest of the manuscript and tables present a quality and a good coherence of the arguments.
Author Response
Comment 1: The introduction could be strengthened by providing more detailed information on the quality and nutritional importance of pitahaya fruit. Additionally, a brief overview of effects of rootstock on internal quality parameters in several species would set the stage for the study's approach.
Response 1: Information about the quality and nutritional importance of pitahaya, as well as, the effects of the rootstock has been included in the Introduction.
Comment 2: The authors should verify that all statistical tests are appropriately applied and that the results are accurately interpreted. The study focuses on statistical differences, however, the authors should better explain the harvest period/rootstock interaction. The results indicate rootstock p-values and harvest p-values.
Response 2: The statistical results were verified. The results of the independent factors (grafting or maturity stage) were eliminated; just the results of the interaction are mentioned in the Results section.
Comment 3: The authors highlight the differences in quality parameters between grafted and non-grafted plants and/or different ripening stages. Could they hypothesize/report what explains these differences?
Response 3: The hypothesis has been included at the end of the Introduction section; and the differences has been explained better in the conclusions.
Comment 4: The studied plants were cultivated in one field collection or in the same pedo-climatic environment? I think it is also important to go more in details on how the cultivation was done, with details on the soil parameters, and fertilization and irrigation regimes.
Response 4: Information about the soil conditions and cultivation management of the experimental plot was added in the Methodology section, in the point 4.1
Comment 5: I think it is important that the authors indicate the fruit harvesting period and the temporal sequence of the sampling.
Response 5: The harvesting period and the sequence of the sampling have been explained in the Methodology section, in the point 4.1
Comment 6: A very important aspect is the preparation of the samples. How much time passed between the harvest and the processing of these fruits? How many fruits were collected and used in subsequent analyses?
Response 6: The sampling preparation and the number of harvested fruits have been mentioned in the Methodology section, in the point 4.1
Comment 7: Since there are many different factors which affect the qualities of a fruit and consequently the profitability of a cultivation, it is necessary to conduct long-term studies to find the best scion/rootstock combination, within the constraints imposed by local climate and economic conditions. I would therefore suggest that the authors repeat the experiments to confirm the results obtained in this research.
Response 7: This suggestion has been included as a recommendation at the end of the conclusions.
Reviewer 2 Report
Comments and Suggestions for Authors
This paper have practical meanings, and can guide the production of yellow pitahaya. The research content and scientific part is relatively less, so the theory of the article is not strong, but still the practical information in this article is also worth reading. There are flaws in this paper, the details are listed below.
1. the title of the paper, is too big and too vague, it should contain specific information of maturity and grafted.
2. table 1, as to the comparative analysis in this study, the color E value (color difference) is necessary,
3. this paper have 2 title of 2.1, please correct it.
4. the P value of data in this paper is kind of to small, usualy p<0.05 is enough, but p < 0.0001 is showed in this paper, please make sure the data processing is correct. meanwhile, p should be italic display.
5. please specific the grading criteria for material maturity, why and how to define the material in 4.1.
Author Response
Comment 1: the title of the paper, is too big and too vague, it should contain specific information of maturity and grafted.
Response 1: The title has been changed, following the reviewer´s suggestions.
Comment 2: table 1, as to the comparative analysis in this study, the color E value (color difference) is necessary.
Response 2: The values for the Delta E (color difference) have been added to Table 1. The formula to calculate this value has been mentioned in the methodology.
Comment 3: this paper have 2 title of 2.1, please correct it.
Response 3: The numeration of the subtitles has been corrected.
Comment 4: the P value of data in this paper is kind of to small, usualy p<0.05 is enough, but p < 0.0001 is showed in this paper, please make sure the data processing is correct. meanwhile, p should be italic display.
Response 4: The statistics were reviewed and the p< 0.05 has been placed only as a footnote in all the Tables. The letter p has been placed in italics in the whole document.
Comment 5: please specific the grading criteria for material maturity, why and how to define the material in 4.1.
Response 5: The criteria for grading the maturity stages of the fruit has been added in the section 4.1
Reviewer 3 Report
Comments and Suggestions for Authors
An excellent investigation with very practical results from both the biotechnological and agronomic point of view. I would only like to point out that the conclusions could be more precise and robust regarding the positive effect of grafts, suggesting to the authors that this relevance be taken advantage of from the title of the manuscript.
Author Response
Comment 1: An excellent investigation with very practical results from both the biotechnological and agronomic point of view. I would only like to point out that the conclusions could be more precise and robust regarding the positive effect of grafts, suggesting to the authors that this relevance be taken advantage of from the title of the manuscript.
Response 1: Thanks for the valuable comment. The conclusions have been rewritten pointing out the positive effects of the grafting.